# Advances in Genetics and Epigenetic Alterations in Alzheimer’s Disease: A Notion for Therapeutic Treatment

**DOI:** 10.3390/genes12121959

**Published:** 2021-12-08

**Authors:** Rubén Rabaneda-Bueno, Beatriz Mena-Montes, Sara Torres-Castro, Norma Torres-Carrillo, Nora Magdalena Torres-Carrillo

**Affiliations:** 1Biology Centre of the Czech Academy of Sciences, Institute of Hydrobiology, 37005 České Budějovice, Czech Republic; 2School of Biological Sciences, James Clerk Maxwell Building, The King’s Buildings Campus, University of Edinburgh, Edinburgh EH9 3FD, UK; 3Laboratorio de Biología del Envejecimiento, Departamento de Investigación Básica, Instituto Nacional de Geriatría, Mexico City 10200, Mexico; bmena@inger.gob.mx; 4Departamento de Epidemiología Demográfica y Determinantes Sociales, Instituto Nacional de Geriatría, Mexico City 10200, Mexico; storres@inger.gob.mx; 5Departamento de Microbiología y Patología, Centro Universitario de Ciencias de la Salud, Universidad de Guadalajara, Guadalajara 44340, Jalisco, Mexico; norma_toc@hotmail.com (N.T.-C.); nora.torres@academicos.udg.mx (N.M.T.-C.)

**Keywords:** Alzheimer’s disease, genetic risk factors, epigenetic mechanisms, older adults, dementia, missing heritability, genome-wide association study (GWAS)

## Abstract

Alzheimer’s disease (AD) is a disabling neurodegenerative disorder that leads to long-term functional and cognitive impairment and greatly reduces life expectancy. Early genetic studies focused on tracking variations in genome-wide DNA sequences discovered several polymorphisms and novel susceptibility genes associated with AD. However, despite the numerous risk factors already identified, there is still no fully satisfactory explanation for the mechanisms underlying the onset of the disease. Also, as with other complex human diseases, the causes of low heritability are unclear. Epigenetic mechanisms, in which changes in gene expression do not depend on changes in genotype, have attracted considerable attention in recent years and are key to understanding the processes that influence age-related changes and various neurological diseases. With the recent use of massive sequencing techniques, methods for studying epigenome variations in AD have also evolved tremendously, allowing the discovery of differentially expressed disease traits under different conditions and experimental settings. This is important for understanding disease development and for unlocking new potential AD therapies. In this work, we outline the genomic and epigenomic components involved in the initiation and development of AD and identify potentially effective therapeutic targets for disease control.

## 1. Background

Alzheimer’s disease (AD) is the hallmark of adult-onset dementia, characterized by progressive and widespread brain damage with massive neuron loss [1,2]. In recent years, our understanding of the factors underlying the onset and development of the disease has improved considerably. However, despite the many resources that have gone into developing effective treatments for the neuropathological changes associated with AD, the disease remains incurable to this day. The consequences of the neuropathology associated with AD are severe cognitive and memory impairments that, in most cases, lead to a progressive reduction in quality of life and life expectancy. These cognitive impairments usually occur in older adults over the age of 65, but in 5 to 6% of patients, the first symptoms appear below this age, sometimes as early as age 30 but more often between 40 and 50 and into their mid-60s. The accuracy of AD diagnosis in clinical practice and research has been questioned following evaluations of several PET studies on amyloid-β-peptide (Aβ). For example, more than one-third of non-carriers of the ε4 allele of Apolipoprotein E (*ApoE*) gene (ApoE4) with a clinical diagnosis of mild or moderate AD have low Aβ levels, and the diagnostic accuracy of the clinical criteria may be limited, not even reaching 77%, with 20–30% of patients having a clinical diagnosis AD that does not match the neuropathological diagnosis [3,4,5]. This is often due to overlap in cognitive testing between non-demented and demented older adults, and in addition, psychiatric comorbidities of various origins may compound the diagnosis of the disease. Overall, then, the diagnosis of AD is increasingly one of exclusion and would require the development of biomarkers in the blood to confirm the correct clinical diagnosis [6]. Although postmortem examination of brain tissue is ultimately required, a number of tools now exist to facilitate premortem diagnosis. Last but not least, vascular dementia and comorbidities associated with type 2 diabetes highlight the strong clinical associations between many vascular risk factors [7].

Long-term mild cognitive impairment (MCI) usually occurs before the onset of the first symptoms of AD and sometimes in the so-called prodromal phase, involving various regulatory, metabolic, biochemical and gene expression processes. Various neuroinflammatory processes may accompany some of the neuropathological changes associated with AD [8]. These changes include progressive deterioration of hippocampal cells and atrophy of the cerebral cortex, which are associated with massive failures of neurosynaptic function and lead to significant cognitive impairment. The changes in synapses and neuronal function are due to extracellular aggregation of Aβ into plaques, which are clearly visible in postmortem brain tissue samples and occur together with distinct patterns of neurofibrillary tangles (NFTs). According to the latest clinical practice guidelines from the National Institute on Aging and the Alzheimer’s Association (NIA-AA), both Aβ peptides and NFTs are characteristic disease-defining factors [5].

Aβ is a 36–43 amino acid peptide formed by the activity of β-site amyloid precursor protein-cleaving enzyme 1 (BACE1) at the Amyloid-β precursor protein (APP) and γ-secretase, both of which catalyze a chain reaction with cleavage activity. The Aβ40 and Aβ42 isoforms are among the major components of amyloid plaques and are important biomarkers for AD. It has been suggested that Aβ43 may also represent an important cargo that is equally important for AD amyloid pathology [9]. The characteristic expression pattern of Aβ in the disease appears to depend on the isoform, as the length of the peptide has been found to correlate positively with the plaque burden and negatively with amyloid deposition in brain vessels. This ultimately implies the development of vascular dysfunction due to amyloid-induced cerebral angiopathy (CAA) [10], but more importantly, the accumulation of peptides in an independent manner in brain regions [9]. The BACE1 gene encodes a transmembrane aspartyl protease with β-secretase activity that processes APP to release its C-terminal fragment (APP-CTF) C99, which is subsequently processed by γ-secretase to release Aβ, a mechanism directly linked to the pathogenesis of AD. Elevated levels of BACE1 have been associated with the deposition of APP-CTFs in human AD brain cells and mitochondria [11,12,13].

Alternatively, nonamyloidogenic processing of APP occurs within the Aβ-domain (APP-Aβ), where cleavage is mediated by enzymes with α-secretase activity that trigger the release of the amino-terminal domain of APP to form APP-CTF C83, as is the case with Disintegrin and metallopeptide domain 10 (ADAM10) in neurons [14]. In this pathway, cholesterol-rich lipid domains appear to play a role in ADAM10 maturation, and cholesterol depletion has been associated with a decrease in α-secretase activity through increased cleavage of the enzyme, resulting in a lower concentration of Aβ-peptides [15]. Inhibition of Aβ production was also associated with increased α-secretase activity of human embryonic kidney cell lines and hippocampal neuronal cells after cholesterol depletion [15,16]. The homologous β-site amyloid precursor protein cleaving enzyme 2 (BACE2) is also an active player in the processing and cleavage of APP-Aβ at the β-secretase site. Recently, it has been shown that mutations in the APP juxta-membrane helix, as well as binding of Clusterin (CLU) to this region, can inhibit the β-secretase activity, which can then promote cleavage of the nascent APP, leading to an exacerbation of disease progression [17].

Early in the progression of AD, activation of central nervous system (CNS) microglia occurs in response to Aβ-aggregation and amyloid plaque deposition in the brain, usually before Microtubule-associated protein tau (MAPT) aggregates into characteristic paired helical filaments and forms NFTs [18]. In the LOAD form of AD, the gene encoding the MAPT protein is crucial because its expression triggers the stabilization of microtubules for cytoskeleton assembly by inhibiting tau aggregation, thereby affecting tau biogenesis and the conformation of NFTs. When phosphorylated tau polypeptides (p-tau) occur at abnormally high levels, they tend to misfold, leading to microtubule destabilization and subsequent neuronal degeneration. These are key factors in disease onset [19] but not the only ones, as they are often preceded by morphological and biochemical changes, many of which are related to mitochondrial function upstream of the amyloid cascade [20], in which Aβ and NFTs are critical for disease progression. With the recent findings from cutting-edge research, the dilemma of whether amyloid plaques are a consequence of progressive pathology or contribute to the pathogenic mechanism has arisen. In contrast to this debate, links between Aβ peptides, NFTs and tau protein biogenesis with pathologies and comorbidities associated with cellular metabolic changes that are at the core of the pathogenesis of AD have been demonstrated with some precision. In mice, for example, impaired cell signaling was found to be closely associated with cognitive impairment [21], while, in the human brain, Meakin et al. identified impaired glucose metabolism as an early feature of AD that is directly related to tau and Aβ metabolism, making diabetes a distinct risk factor for the clinical onset of AD [7]. Thus, the question of which mechanisms are responsible for NFTs formation and neurodegeneration due to Aβ deposition needs to be answered urgently [22].

It is well known that the genetic component usually explains only part of the causal factors in AD. Therefore, it is important to better understand the functional consequences of these genetic associations. To this end, research on the epigenetics of neurological disorders has become particularly important in recent years to alleviate some of the limitations of earlier candidate gene studies, which tend to explain the genetic variance almost exclusively. Advances in microarray and genome sequencing technologies, such as high-throughput next-generation sequencing (NGS) and genome-wide association studies (GWAS), have encouraged genome-wide cohort studies to gain further insight into the epigenome and the factors that cause genetic variance in disease phenotypes [23]. Combined with large association studies of the epigenome (EWAS), these new resources have proven more effective in identifying multiple causal factors associated with AD, including genetic variations and epistatic effects of gene–gene interactions [24]. Here, we review the key findings on genetic factors and novel epigenetic mechanisms underlying the development and pathogenesis of AD. This includes insights into the current treatments in development and promising therapeutic targets.

## 2. The Genetics of AD

The inheritance of AD depends on the interaction of several genetic factors whose expression is the result of complex biochemical pathways and cellular communication that occur prior to the formation of Aβ deposits and NFTs [25]. Two forms of AD are distinguished according to the age at which the disease first appears: a genetically rare familial form, which accounts for less than 6% of all patient cases, of which about 60% have a family history of AD and 13% of which have an autosomal dominant inheritance, and a multifactorial sporadic form, in which genetic factors and environmental influences may determine susceptibility AD. The rarer familial form usually occurs at an age of less than 65 years as early onset AD (EOAD), usually in the thirties or forties of the affected individual. The prevalence of neurodegenerative dementia associated with EOAD is characterized by rapid disease progression and shorter estimated survival [26], which worsens as patients age and reach the age group of 65 years [27]. This has been linked to the formation of phosphorylated tau protein (p-tau) deposits at a very early stage in the brain [28]. The sporadic form, which is the most common, usually occurs as a late onset AD (LOAD) around the age of 65 and is genetically more complex, both in terms of inheritance and etiopathology [29,30,31].

### 2.1. Important Susceptibility Genes

When it comes to the genetic study of complex human traits such as AD, a variety of approaches have been used, ranging from genetic association analyses to candidate gene screening. Anticipating advances in genome-wide analysis, candidate gene studies conducted in AD patients and control subjects have compared the frequency of genetic variants and identified the protein-coding genes of the amyloid-β precursor protein (*APP*), and the subunits PS1 and and PS2 of presenilins (*PSEN1*, *PSEN2*), and *ApoE* [32,33] as important susceptibility factors for AD. Mass association studies based on tools such as GWAS have enabled the discovery of new unknown risk variants. The initial phases of this research identified more than 25 single nucleotide polymorphisms (SNPs) and numerous variants of these risk genes that contribute to the cumulative risk of AD (Appendix A).

The most common allelic form of *ApoE* is ε3 (ApoE3), which contains a cysteine residue at 112 and an arginine at 158, whereas the ε2 allele (ApoE2) with corresponding cysteines at both positions is the rarest form. In between, the ε4 allele (ApoE4), characterized by arginine residues at the same positions, is the predominant risk and causative factor for the sporadic form of AD, while it is a genetic risk only in EOAD [34]. The association of *ApoE* alleles and genotypes and aging was examined in detail in a total of 10,623 participants from several case–control and cohort studies of the European population. The subjects had no evidence of AMD (age-related macular degeneration) and indistinguishable AD status, i.e. the reported frequencies actually corresponded to normal frequencies observed in the population. The results of this study showed that the allele frequencies of ApoE3, ApoE2 and ApoE4 were 74.98%, 8.62% and 16.40%, respectively, in subjects younger than or equal to 65 years of age and 78.74%, 7.74% and 13.52%, respectively, in subjects above this age [35]. In the case of the ε4 isoform, the substitution of several amino acids leads to a structural change that favors binding to VLDL (very low-density lipoprotein)-type particles and to HDL (high-density lipoprotein) in the case of the ε2 and ε3 isoforms. The neuroprotective mechanism of ε2 is due to its structural affinity for HDL, resulting in individuals with the ApoE2 genotype having higher HDL levels [36] and also better overall levels of Insulin-like growth factor I (IGF-I) and -sensitive Glucose transporter type 4 (GLUT4) signaling compared to ApoE3 and ApoE4 [37]. In contrast to the ApoE2 allele, which is protective, particularly in women [20,37], ApoE4 is associated with a dose-dependent increased risk of AD. A single ApoE4 allele has a two- to three-fold increased lifetime risk, while the ApoE4/ApoE4 genotype significantly elevates the risk of developing MCI and AD over the single genotype. The risk is also age-dependent. Individuals with the two ApoE4 alleles are, on average, 65 years old (50% males and 60% females), and those with the ApoE3/ApoE4 genotype are 85 years old (23% males and 30% females), but there are no significant differences between the sexes [20,34]. ApoE4 has an estimated heritability (SNP-based) of 0.13–0.33 and about 25% of the overall heritability (LOAD), but these estimates tend to increase significantly in twin studies [38,39,40].

In familial AD, autosomal dominant variants of *PSEN1* and *PSEN2* and Aβ precursor genes are responsible for 5–10% of the EOAD cases, while the additive effects of different polygenic variants may explain the remaining cases of AD [41]. Regardless of the location of these genes, which include APP, on physically separate chromosomes, they are expressed through common biological pathways with Aβ metabolism, so mutations in these genes have great potential to influence the amyloid pathogenicity and early onset [42]. For example, a missense variant of *APP* at codon 673 increases the risk of AD [43], and a mutation in *PSEN1* alters the processing of APP and promotes the accumulation of Aβ plaques [40]. In addition, mutations in the Aβ sequence of *APP* can promote fibrillation and early cognitive impairment or cause inhibition of Aβ-peptide that prevents neuronal dysfunction [44].

The second important risk locus is considered to be Bridging-integrator 1 (*BIN1*), which is now known to have a synaptic function that can alter neuronal degeneration and potentially trigger NFT-related diseases [45]. Some polymorphisms are involved in tauopathy but not β-amyloid [46], and an increased expression suggests an increased risk of memory impairment, which promotes disease progression by modulating tau pathogenesis in AD patients [45,46,47]. As seen when comparing AD patients and control subjects, differential gene expression also occurs at other loci in different brain regions that are associated with amyloid pathologies to varying degrees. For example, the gene encoding the α1 antichymotrypsin protein of Serpin family A, member 3 (*SERPINA3*) is an inhibitor of serine protease enzymes that has been associated with increased dementia risk [48] and is upregulated in the brain of AD patients [18,49], which increases p-tau levels and promotes Aβ deposition and formation of NFTs and can lead to neuronal death [50,51,52].

There are other first-order risk factors for AD, but their mutational effects are less predictable than those of *ApoE* because of their lower penetrance. These include genes encoding proteins that increase the risk for LOAD, such as Phosphatidyl-inositol-binding clathrin assembly (*PICALM*), Apolipoprotein C-1 (*APOC1*), Sortilin-related receptor 1 (*SORL1*), Complement receptor 1 (*CR1*), ATP-binding cassette member 7 (*ABCA7*) and Estrogen receptor 1 (*ESR1*) [40,53,54,55,56,57]. Several GWASs and meta-analyses have also found LOAD risk variants associated with microglial function and neuroinflammatory processes in the Caucasian population, such as *CLU*, Myeloid surface antigen (*CD33*), Membrane domain 4 subfamily member A (*MS4A4*) and CD2-associated protein (*CD2AP*) [58], but studies in Asian populations have mostly yielded mixed results, generally showing weak or no associations with the same genes. Of these genes, *CLU* has emerged as the third-most genome-wide significant risk locus for LOAD and is an important component of disease progression from MCI to LOAD [59]. *CLU* is notably regulated through interactions with other risk loci involved in various regulatory processes of tau pathology, such as Aβ clearance, Aβ binding and deposition [60,61] and cerebral neuroinflammatory stress responses [62,63,64]. Ultimately, *CLU* expression alone or in concert with other loci leads to neuronal dysfunction through fibrillation and amyloid plaque formation [1,65,66].

NGS, whole-genome (WGS) and whole-exome (WES) sequencing-based studies have further documented several associations between genetic variants and AD, such as between Triggering myeloid cell receptor expressed-2 (*TREM2*) and EOAD and LOAD forms in Caucasian populations. A number of rare variants have been found to increase the risk of LOAD two to four-fold by a factor comparable to that of an ApoE4 carrier [67,68]. The best-known *TREM2* variant is R47H, the origin of which is the rs75932628 polymorphism encoding a substitution of arginine for histidine at amino acid 47. The risk for LOAD is significantly increased in carriers of the R47H variant, with a more rapid onset of symptoms and more rapid impairment of cognitive abilities. This association exists predominantly only in Caucasian populations but not Asian, and in many cases, the effects of the R47H variant appear to be related to amyloid pathology and NFTS formation. Studies conducted in vivo and in vitro have yielded conflicting results regarding the involvement of *TREM2* in Aβ uptake by microglia [69,70], but higher levels of the tau protein (total and p-tau) have been found in the CSF of individuals who are carriers of the R47H variant compared to individuals who are not carriers [71]. *TREM2* and *ApoE* have also been proposed to jointly influence the pathogenesis of LOAD, although the mechanisms of interaction between the two risk loci are still unknown [62,72].

In a recent NGS study, Giau et al. analyzed familial EOAD samples from 67 subjects in an Asian population with a panel of 50 genes, including causal variants and putative causal variants implicated in neurodegenerative diseases. Three nonsense mutations in *PSEN1* (G209A, G417A and T119I) and one identified *PSEN2* variant (H169N) were detected in 6% of patients. In addition, 67 nonsense mutations were identified in LOAD susceptibility genes possibly involved in cholesterol transport, neuroinflammation and the modulation of Aβ pathogenesis and amyloidosis. They found another 70 novel nonsense variants in other genes, including *MAPT*, *PRNP*, *CSF1R* and *GRN*, which have been linked to various neurodegenerative diseases. The findings of these authors suggest that other genes associated with neurodegenerative diseases should be investigated in addition to the clinical diagnosis of AD [73].

The *CD2AP* gene is another important genetic factor contributing to the risk of AD, although its function in the brain is not well-understood. Since it is involved in maintaining the blood–brain barrier, both the inactivation and loss of function of *CD2AP* are thought to play a critical role in the pathogenesis of AD [74], which is consistent with the recent discovery of EOAD-associated variants [75]. In a recent study, Yan et al. used the SNaPshot Multiplex System primer extension screening technique (Applied Biosystems, Foster City, CA, USA) to examine the association between multiple SNPs and the susceptibility to AD in a sample of 215 patients with AD and 205 controls in a sex- and age-mixed cohort. Subsequent sequencing tests have revealed a positive association between the G allele of the *FERMT2* gene polymorphism rs17125924 and the risk of developing AD, which was increased in individual AD with ApoE4. Specific genotypes of some *CD2AP*, *PTK2B* and *HLA-DRB1* gene polymorphisms were also associated with their expression levels and the likelihood of developing EOAD [76].

Together with the new NGS analyses based on genetic screening in specific regions of a gene, detecting risk variants in the brain of patients with AD becomes possible by tracking the regulatory activity of gene expression. For example, previous studies have suggested an association between variants of the gene *MAPT* and the risk of LOAD [77,78], although the results have been inconsistent in many cases. However, more recent results based on gene haplotypes have shown a strong influence of the regulated expression of the gene in the brain [79]. Similarly, several association studies have shown mixed results regarding the association of *TOMM40* (outer mitochondrial membrane translocase 40) with AD and the dysregulation of mitochondrial function independent of ApoE4 [80,81], but a recent haplotype study showed differences between *TOMM40* haplotypes encoding ApoE4 and ApoE3 in the association probability for AD [82].

### 2.2. Missing Heritability: Epistasis Explains Undetected Associations in Risk Loci

As with many complex human traits, it should be noted that a large number of the genetic risk factors identified to date that are associated with AD appear to explain only a minimal portion of the heritability of AD [83]. One explanation for this “missing heritability” is thought to be the existence of rare and undetected variants that contribute greatly to the amplification of the disease phenotype, such as the occurrence of mutations in *ADAM10* [84]. Therefore, it is not always easy to find a link between the genetic risk factors and AD, despite the increasing research aimed at providing information on the pathogenesis of the disease. For example, one of the largest GWAS-based meta-analyses published to date on AD showed no genome-wide significance for the myeloid cell surface antigen (*CD33*) and Desmoglein 2 (*DSG2*) genes [85], whereas *CD33* was later found to have a strong association with an increased AD risk in Caucasian and Asian populations [86,87]. Additionally, the *ATXN1*, *APOC1*, *TRPC4AP*, *PLXNA4* and *CUGBP2* genes and other uncharacterized loci on chromosome 14q31.2 reported in family-based GWAS studies failed to be reproduced by case–control approaches [88,89,90].

Another explanation that has attracted considerable interest in recent years is that, as in other complex diseases, the susceptibility toward AD is determined by additive effects and epistatic interactions of multiple genetic variants, consistent with the fact that AD and genetic transmission are inherently multifactorial. For example, *BIN1* has been found to interact with coding products of similar functional pathways affected in AD, such as *PICALM*, *MAPT*, *ABCA7* or *SORL1*; *CLU* and *ApoE* have additive effects on lipid trafficking and affect Aβ deposition and *ABCA7* interacts with the major risk candidate genes *BIN1*, *CLU* and *PICALM* [30] (the epistatic interactions between these and other coding protein genes are shown in Figure 1). The genetic interactions between the rs670139 and rs11136000 polymorphisms of the *MS4A4E* and *CLU* genes predict up to an 8% increased risk for the occurrence of AD [91]. Thus, the epistatic dominance effect of *MS4A4E-CLU* is already among the major risk loci identified to date, along with *ApoE*, *APP* and *TREM2*. Moreover, the *CD33* and *MS4A6A* risk variants play a pivotal role in amyloid pathology, as they are responsible for regulating *TREM2* expression in CSF [92,93,94,95].

Currently, the use of updated multivariate GWAS (MGAS) can provide better results in the discovery of new susceptibility genes and also a better understanding of the interactions between already identified risk genes. This type of analysis opens up new possibilities in the study of complex traits by using information on biological relationships in combination with bioinformatics tools such as a network analysis of protein interactions (PPI) [96,97] (an example of a PPI network analysis is depicted in Figure 1). Last year, for example, Meng et al. found known risk factors such as the *ApoE* and *APOC1* genes after applying multivariate screening based on the extended Simes method (GATES), together with a PPI network analysis, to eight subcortical measures of the AD neuroimaging phenotype, and also discovered novel genetic variants (*LAMA1*, *XYLB* or *NPEPL1*) with potential influence on the disease [98]. In another study, analyzing the expression profiles in combination with PPI networks in hippocampal tissues from 76 AD patients and 40 healthy controls, it was found that a total of 80 genes were differentially down- or up-regulated. These are genes that affect processes such as neuronal signaling and synaptic transmission functions of GABAergic circuits in the CNS, all of which are associated with AD pathology. The researchers also found other candidate genes potentially relevant to the progression of AD and as biomarkers for early diagnosis, such as *ITGB5*, *RPH3A*, *GNAS*, *THY1*, *NEK6*, *JUN*, *GDI1*, *GNAI2*, *ERCC3* and *CDC42EP4* [99]. Other gene-based multivariate tests, such as the Versatile gene-based assay (VEGAS) or Multiphenotypic association analysis (MultiPhen), can also provide additional data on susceptible brain regions of AD-related functional areas [100].

## 3. Alzheimer’s and the Epigenome: Filling the Gap?

Epigenetics is concerned with changes in DNA that can disrupt gene expressions and phenotypes without altering the nucleotide sequences. More generally, epigenetic changes can be understood as any mechanism by which the environment can alter the phenotype without altering the genotype and may require a signaling cascade from the transcription factor expression [101]. Currently, more than twenty epigenetic mechanisms are known, including chemical marks on DNA, genomic imprinting, noncoding RNAs (ncRNAs), post-translational modifications of histones (PTM-Hs) and a number of confounding factors related to environmental modulations, to name a few (the main epigenetic mechanisms of AD are illustrated in Figure 2). All of these mechanisms are important players in the process of transcriptional regulation and can promote functionally relevant changes in the gene expression and cellular identity [102], control of the transcription-binding factors, utilization of alternative transcription start sites and splicing processes [103,104,105] and the regulation of development through gene activation or silencing [106,107].

Consistent with the above limitations, epigenomic studies may provide an entirely new approach to investigate associations between disease phenotypes and other previously overlooked non-genetic factors. In this sense, epigenetic mechanisms would act as intermediaries of environmental factors and genetic risk components throughout an individual’s life [108,109]. Genetic and epigenetic information are not independent but complementary and can theoretically provide relevant information on the causality of DNA sequence variations. For example, epigenetic variation within a given genomic locus may be a direct consequence of DNA sequence variation, with epigenetic modifications being directly linked to environmental influences on phenotypes and potentially regulating or altering the expression of genetic variants through environmental modulation [110,111]. However, as we mentioned earlier, this dichotomy poses an additional problem, because the heritability of complex traits is low. Therefore, the transmission of such traits between generations is thought to be due to non-genetic changes, further obscuring the link between genetics and epigenetics and misinterpreting the variations underlying these mechanisms. Whether or not this is the case, epigenetic changes are a crucial factor that can explain the non-genetic component associated with the lack of heritability of complex traits. How epigenetic variations affect individual phenotypes, gene expressions and brain developments in the context of AD should be understood from the perspective of inheritance across generations without changes in the DNA sequence [30,112].

### 3.1. Epigenetics Alterations of AD

As mentioned earlier, interactions between genetic and environmental changes play a crucial role in the etiology of sporadic AD. For example, age and other factors indicative of healthy or unhealthy lifestyle habits, such as diet, smoking and education level, have been associated with the occurrence of AD [113,114,115]. Epigenetic mechanisms most commonly associated with AD include DNA methylation, PTM-Hs, and gene silencing of ncRNAs. These generally do not occur in isolation, but rather in a complex interplay in which these modifications interact to regulate various aspects of genomic and trnaskriptomic domains that may ultimately influence important cognitive processes such as memory and learning ability [116,117]. For example, there is evidence that mutations in active genes can be altered by neuronal activity and disrupt the modulation of epigenetic pathways that trigger disease states associated with different neurological disorders [118,119]. Despite the association between many of these factors and the occurrence of epigenetic changes during the development of AD, the process that drives the interaction between each factor and leads to different rates of disease progression is not well-understood. For more information, see several studies on the epigenome in AD (Appendix A).

#### 3.1.1. DNA Methylation

By its eminently repressive action, DNA methylation favors the attraction of proteins known to be involved in gene silencing [30] through the action of DNA methyltransferase (DNMT) enzymes that catalyze the transfer of methyls from S-adenosylmethionine (SAM) to 5′-cytosine terminals (5mC), which are then bound by phosphodiesterase to guanines in CpG dinucleotides. Therefore, coding genes contain promoter regions that are often highly enriched in CpG, forming CpG islands (CGI) with characteristic elevated methylation profiles (Figure 2).

Ten eleven translocation enzymes (TETs) catalyze the synthesis of 5-hydroxymethylcytosine (5hmC) from 5mC and represent another important epigenetic modification. Apparently, the concentration of 5hmC depends on the enzymatic activity of TETs, which, in turn, is related to the concentration of Aβ and tau hyperphosphorylation, so that the concentration of 5hmC is directly related to the stage of AD and its role in the pathogenic progression of the disease. There is evidence of a relationship between 5hmC concentration, age and gene regulation in neurodegenerative processes [120,121], possibly due to the dysregulation of 5hmC during brain development [122]. Therefore, in postmortem samples of human AD brains, the reduction in TET activity might parallel the reduction in 5hmC concentration, as observed in neuronal tissue from mouse models of AD [123].

The association with a suppressed gene expression highlights the importance of CGI regions in gene regulation. However, as they are generally not methylated, this also suggests that their gene regulatory role is not solely dependent on DNA methylation. Thus, many DNA methylation studies have revealed tissue-specific methylation analyses in a small number of brain regions [124], and it is important to consider not only the DNA methylation status but, also, the specific environment in which it occurs [103]. The need to clarify whether these epigenetic changes affect specific or general DNA regions and whether the overall trends of decreasing and increasing DNA methylation are consistent raises several questions that have been addressed by different approaches to date with varying results, largely due to the large differences in the data and experimental setups used. For example, the occurrence of the Aβ peptide has been associated with a general state of genomic hypomethylation in cell lines [125], although elevated levels of DNA methylation have also been reported in mouse models and in studies using postmortem samples of human brain tissue [126]. Early studies focusing on alterations of amyloid hypothesis gene loci have not found sufficient evidence for a general pattern of the disease. This apparent inconsistency could be facilitated by changes in methylation profiles between affected regions or between individuals under experimental changes [119]. In addition, differences in methylation in AD brains could be relatively small and associated with specific loci in DMRs [85,127].

In recent years, a link between DNA methylation and AD has been repeatedly demonstrated, and the first evidence of this comes from two studies using human brain and blood samples. In the first study, the researchers analyzed the DNA of dorsolateral PFC from brain biopsies of AD patients and identified amyloid load-dependent DMRs of the genome, many of which were associated with neuropathology, including multiple methylation sites in the AD susceptibility loci *BIN1* and *ABCA7*. They also performed the validation of eleven DMRs in a new cohort of individuals and performed RNA expression studies that identified seven genes with altered expressions: Anchyrin 1 (*ANK1*), Disc-interacting protein 2 homolog A (*DIP2A*), Cadherin 23 (*CDH23*), Rhomboid protein 5 homolog 2 (*RHBDF2*), 60S ribosomal protein L13 (*RPL13*) and Serpin family members 1 (*SERPINF1*) and 2 (*SERPINF2*). Interestingly, the methylation changes appeared to occur early in the disease, as evidenced by the fact that these changes reappeared in patients with characteristic amyloid pathology, even if they had not yet developed cognitive impairment [128]. In a second study, genome-wide methylation variations in the human genome were analyzed using brain and blood samples from different regions in four independent cohorts. The researchers found methylation hotspots in the *RHBDF2*, *RPL13* and *CDH23* genes, which were consistent with the results of the previous study and provided evidence for DMRs in different brain regions. *ANK1* hypermethylation in the cerebral cortex was correlated with AD-associated neuropathology in the brain. These significant methylation changes in the cerebral cortex contrasted with the absence of methylation in the cerebellar region protected from neurodegeneration. In contrast to the postmortem blood samples, most DMRs were present in the premortem blood samples, and the presence of multiple methylation sites in nearby genes was associated with changes in the disease expression [129].

Although genomic regions with variable methylation patterns (DMRs) have been observed to overlap with AD, recent research has yielded conflicting results. On the one hand, abnormal levels of DNA methylation have been found in AD patients [130], and on the other hand, some studies concluded that there were no significant differences in the levels of the epigenetic methylation marker SAM between AD patients and healthy individuals [131,132]. In a pyrosequencing study, researchers compared the DNA methylation profiles of several AD-associated CGI genes in the cerebellum, inferior temporal gyrus and superior parietal lobe between individuals with and without dementia. They found evidence that altered DNA methylation leads to changes in gene expression, representing a possible influence of DNA methylation on the phenotype AD. However, while DNA methylation profiling showed significant differences for *MAPT*, *APP* and *GSK3B*, this was not the case for the *ApoE*, *PSEN1* or *BACE1* genes [133]. Foraker et al. apparently shed light on differential DNA methylation with respect to the *ApoE* gene. In their study, they found that there was tissue specificity in methylation profiles, with two AD-specific DMRs associated with CpG islands (CGIs) of *ApoE*. The methylation of CGIs was decreased in AD brains, while, in ApoE4/ApoE4 controls, the presence of the ApoE4allele was associated with increased methylation and the main differences in the DMRs were between AD and subjects with the ApoE3/ ApoE4genotype [134]. These results suggest differential methylation in subpopulations of brain cells with CGI specificity.

Complementing this, recent research has provided results that are consistent with earlier findings. For example, Yu et al. examined the DNA methylation patterns of about 30 loci in more than 700 autopsies from AD pathology patients aged 66–108 years. About 60% met the diagnostic criteria for AD pathology associated with BIN1 methylation in the brain [135]. When the trait AD was replaced by two disease-specific quantitative and molecular traits, the Aβ load and NFTs density, the association was consistently stable. In addition, the expression of BIN1 was found to be directly or indirectly related to the expression of the *HLA-DRB5*, *SLC24A4*, *ABCA7* and *SORL1* loci [39], and in a previous study, the expression of the RNA transcripts of *ABCA7* and *SORL1* was also dependent on the NFTs density [135]. Recently, the hypomethylation of *BIN1* was shown to be strongly associated with preclinical AD disease. Using 330 and 484 subjects with and without Aβ-related pathology, such as cognitive impairment, the researchers analyzed whether *BIN1* methylation levels in peripheral blood were associated with susceptibility AD and with early-stage changes in LOAD cerebrospinal fluid (CSF). The authors found that subjects with AD traits had a characteristic hypomethylation status in BIN1, independent of the ApoE4 genotype, and also showed higher p-tau levels and lower CSF levels of an important AD biomarker, Aβ42, compared to the control group [136].

*ABCA7* is primarily responsible for the lipid efflux from cells into lipoprotein particles but also has a regulatory effect on APP processing, secretion and clearance [137]. Recently, Smith et al. performed an epigenomic association study on samples from the superior temporal gyrus and prefrontal cortex (PFC) of 147 subjects to discover DMRs known to be associated with the neuropathology of AD. They replicated their findings in two independent datasets (*n* = 117 and 740). In blood samples from patients with AD and the control subjects, as well as in individuals with MCI, elevated levels of DNA methylation in the homeobox cluster A (*HOXA*) and homeobox B6 (*HOXB6*) regions were found to correlate with neuropathology (Figure 2), highlighting the role of epigenetic variations within the homeobox gene family as a potential AD target for future research [138,139].

#### 3.1.2. Histone Modifications

Posttranslational modifications are the second relatively important epigenetic mechanism associated with AD. The core histones are involved in the conformation of nucleosomes, whose position and compactness are influenced by the DNA sequence, while their regulation is mainly through covalent histone modifications occurring at the N termini. The diversity of PTM-Hs includes methylation, acetylation, O-GlcNac modification, ADP-ribosylation, the phosphorylation of serine and tyrosine residues, ubiquitination and the binding of molecules of the Small ubiquitin-like modifier class (SUMO) system via SUMOylation.

The methylation of lysine, serine or arginine by the enzymatic action of histone methyltransferases (HMT) and demethylases (HDMT) and their acetylation by acetyltransferases (HAT) and deacetylases (HDAC) are among the best-studied epigenetic events, with modifications controlled by fluctuations in the total histone levels and the enzymatic activities of several functionally distinct histones [140] (Figure 2). Each modification may be associated with activation of the target gene, as in lysine acetylation [141,142,143,144] and in other cases with repression and inactivation of a gene [145]. Since the significance of each modification depends on the amino acid modified and the number of methyl groups added, and amino acids are susceptible to modification by different enzymes, the “histone code hypothesis” states that histone modifications can be specifically conjugated to form a functional complex capable of directly regulating a chromatin structure and altering nucleosome formation.

These regulatory pathways are particularly important in the progression of AD, especially since research has shown that histone acetylation is associated with disease development. Previous evidence showed a co-factorial relationship between AD and decreased histone acetylation, although more recent studies have reported increased acetylation levels and genomic region specificity [146]. For example, the epigenetic mark H4K16ac of H4 tends to become more acetylated with age in healthy individuals, whereas the mark tends to disappear in regions of the genome near age- and AD-dependent genes. Nativio et al. found that peak redistribution of histone acetylation (H4K16ac) significantly increased and decreased in postmortem samples of the lateral temporal lobes of AD individuals compared with controls, which included cognitively healthy elderly and young subjects. They also observed that the regions with AD-associated H4K16ac alterations were enriched with single nucleotide polymorphisms [147]. Further studies have shown increased activity of the enzyme HDAC2 in the brains of AD subjects [148], although downregulated histone marks were also observed in quantitative states of H3K18/K23 acetylation by a liquid chromatography–tandem mass spectrometry (LC–MS/MS) analysis based on selected reaction monitoring (SRM) [149]. On the other hand, the inhibition of HDAC6 enzymatic activity suppresses vital functions in the mitochondria of hippocampal neurons induced by Aβ [150,151], while the increased deacetylation of α-tubulin and tau protein by HDAC6 may promote disease slowing by supporting microtubule stabilization and assembly [152].

The CREB-binding protein (CBP) plays a key role in the pathogenesis of tau in AD brains through its acetyltransferase activity, which is supported by the involvement of lysine acetylation of the *MAPT* gene in the formation of amyloid plaques [153,154]. During the processing of APP, the intracellular domain (AICD) (formed together with an extracellular amyloid fragment) is induced to interact in vitro with the histone acetyltransferase TIP60 HAT protein complex and act in tandem as a co-transcriptional activator. The results in neuronal cultures have shown that mutations in *PSEN1* inhibit the degradation of the transcriptional coactivator CBP from the HAT protein by the proteasome, resulting in an increased gene expression [155]. Alternatively, p300 acetyltransferase activity suppresses the degradation of p-tau, thereby regulating tau acetylation, which promotes fibrillation and, ultimately, neurodegeneration by destabilizing the microtubules and disrupting the cytoskeletal assembly [156]. Both CBP and p300 are also involved in the selective Aβ-induced acetylation of various lysine residues of nuclear transcription factor kappa B (NF-κB), and these signaling pathways may play a role in the activation or inhibition of NF-κB-mediated inflammatory responses [157]. Other studies in various animal models found that decreased histone acetylation is associated with the development of AD. In in vitro models, NF-κB was strongly regulated by the enzymatic activity of SIRT1, which is also involved in the deacetylation of tau [158,159] (Figure 2). Moreover, an in vivo study in mice has shown that overexpression of the *SIRT1* gene provides protection against neurodegeneration of AD, although it is unclear whether the gene’s coding product acts through epigenetic mechanisms and/or in conjunction with other genes [160]. A second important substrate of SIRT1 activity is the tumor suppressor transcription factor p53, which is also altered by p300, leading to abnormally high acetylation levels in AD brains [161].

Several neurodegenerative diseases, including AD, share an important factor in their development, namely the covalent binding of ubiquitins to lysines in a process known as ubiquitination of proteosomes. The mechanism of ubiquitination occurs through the action of E-type enzyme complexes that activate, conjugate, and bind to various functional domains, with the E3 subtype having the greatest number of enzymes described and the greatest specificity of binding sites. Ubiquitin is actively involved in proteasome protein degradation, but also plays roles as diverse as the control of apoptosis, autophagy, cell cycle, transcriptional regulation, and regulation of intercellular signaling cascades related to the DNA repair system. Song et al. found that the activity of the enzyme E2–25K/Hip-2 was upregulated in neurons exposed to Aβ42 and mediated Aβ amyloid-dependent neurotoxicity [162]. In subsequent studies in 3 × Tg AD mice, Oddo et al. observed a correlation between Aβ accumulation and proteasome function affecting tau pathology. The injection of anti-Aβ antibodies into the brains of mice resulted in Aβ clearance and a significant reduction in early, but not late, tau deposition, such that the inhibition of proteasome activity prevented the degradation of aggregated tau and promoted its accumulation [1,163]. Therefore, impaired proteasome function would lead to increased Aβ and tau concentrations, which are directly related to the pathology of AD.

#### 3.1.3. Noncoding RNAs

Noncoding RNAs represent a relatively understudied epigenetic feature associated with Alzheimer’s disease pathology, but they are essential for the proper functioning and constitution of cells. They can take the form of infrastructural mRNAs, such as small fragments of nuclear (snRNAs), nucleolar (snRNAs), ribosomal (rRNAs) and transfer RNAs (rRNAs), as well as being part of ribonuclease enzymes and RNA telomerases (e.g. RNase P and TERs) [164]. Longer nRNA molecules form microRNAs and Piwi-interacting RNAs with sizes between 22 to 23 nt (miRNAs) and 26–31 nt (piRNAs) and medium and large ncRNAs between 50 and 200 nt. Their expression occurs in a variety of genomic regions important for processing APP, Aβ formation and the regulation of neurodegenerative processes [165], and their regulatory function depends on the developmental stage and cell differentiation, as well as on various environmental and external stimuli.

#### 3.1.4. MiRNAs

MicroRNAs belong to a category of endogenous small ncRNAs that directly or indirectly regulate posttranscriptional gene expression by inhibiting the transcription or degradation of mRNA. In a first step, miRNAs are transcribed as pri-miRNA molecules with a 5′ cap at 5′ and a polyadenine (poly-A) tail at 3′ and subsequently converted into pre-miRNAs characterized by short loop structures of 70 nucleotides by the action of the RNase III Drosha–DGCR8 complex. In the cytoplasm, the miRNA precursors are matured into miRNAs by the ribonuclease Dicer and initiate the production of the silencing RNA-induced complex (siRC). Normal chromatin function can be severely impaired when the miRNA–epigenetic regulatory circuit is disrupted, ultimately leading to various neurodegenerative diseases [166,167]. SNP-based dysregulations of miRNA expression levels in pre-miRNA or mature miRNA are thought to mediate the epigenetic status of AD, and there is growing evidence that they are involved in LOAD pathogenesis [145,168,169,170,171]. For example, downregulated miRs-221, 144 and 374 expression levels have been found in the brains of LOAD patients compared to healthy controls [171,172], and miR-221 has also been found in serum samples from AD patients suffering from amnesia [173], suggesting that circulating microRNAs may be an important factor in the progression of AD.

Schonrock et al. found that several miRNA molecules such as miR-9, miR-137, miR181c and miR-29a/b were downregulated in hippocampal cell cultures bridging the biomarker Aβ42 deposition in mice [168]. In addition, there is evidence from a recent meta-analysis that miR-129 is able to regulate several target genes involved in synaptic plasticity, including the *CAMK4* gene, which encodes the Ca/M-dependent protein kinase IV, which is present at low levels in hippocampal cells from AD brains [174]. Interestingly, CaMKIV is an activator of CBP, which is responsible for regulating synaptic functions in neuronal cells, and its inhibition may promote amyloid- and tau-associated pathologies [154,175,176]. The inhibition of Dicer or Drosha enzymes leads to the cessation of miRNA biogenesis, which indirectly affects methylation patterns. Dicer-deficient mouse ESCs lacking the miR-290 cluster are associated with the downregulation of DNMT1, DNMT3A and 3B, leading to a decrease in DNA methylation. This family of miRNAs targets Retinoblastoma-like corepressor protein 2 (RBL2), which inhibits the transcription of DNMTs [177,178]. Similarly, miRNAs exert a wide range of effects to alter the amyloidogenic processing of APP to neurotoxic Aβ42, which is influenced by the sequential actions of the cleavage enzymes BACE1 and γ-secretase (the mechanism of action of miRNAs is shown in Figure 3).

Recently, Jain et al. found a high expression of miRs-27a-3p, 30a-5p and 34c in the CSF of AD dementia patients. A combined analysis of this miRNA pattern, previously associated with neurodegenerative disorders such as those leading to memory impairment, with measurements of p-tau and the Aβ-42/40 ratio was also able to correctly label and unambiguously diagnose patients with AD dementia in 98% of cases and distinguish them from healthy controls [169]. Despite these findings, and although the biologically relevant role of several miRNAs in the diagnosis, progression and therapy of AD has been recognized, no universally accepted miRNA has yet been identified for use as a clinically relevant biomarker.

#### 3.1.5. Piwi-Interacting RNAs

PiRNAs are short regulatory ncRNA sequences with a methyl group at their 3’ end and predominantly uridine at their 5’ end, which gives them stability. They are synthesized from intergenic repeat regions, similar to any sRNA, except that the newly produced piRNAs bind to a specific PIWI protein of the Argonaute protein family, whose characteristic endonuclease activity is determined by protein regions of the RNase H family. This endonuclease activity enables the regulation of transposons, so that piRNAs are associated with genome maintenance through transposon repression [179]. However, their most important functional role is gene silencing in animal germ cells [180], which has attracted attention, because they can act as transposon inhibitors against uncontrolled transposable elements, leading, for example, to imbalances in the genome and tumor development [181]. Their role in complex human diseases, especially those in which multiple elements interact, such as AD, remains to be elucidated.

Recent findings showed that piRNAs can exhibit a different expression pattern in AD brain tissues. For example, Qiu et al. identified the differential expression of about 100 piRNA molecules in PFC, while no significant differences in their expression were found in the healthy controls. In the same study, up to 150 piRNAs were found to be expressed in AD-affected human brains, with most of them upregulated (146) and only three downregulated. Some of these piRNAs were selected as potential AD-specific markers based on their elevated expression levels [182]. For example, this team found that the molecules piR-61646, piR-31038, piR-33880, piR-34443 and piR-37213 were upregulated 10-fold, making these piRNAs a fairly reliable AD signature, especially compared to other piRNAs [183]. In another recent study, three piRNAs were identified in the CSF of AD dementia patients, where the expression of piR-019324 was downregulated, while the expression of piRNAs 019949 and 020364 was upregulated. Moreover, the combined analysis of the piRNA expression patterns from p-tau and the Aβ-42/40 ratio measurements served not only to detect AD dementia but to also predict its progression from the MCI stage [169], making piRNAs, together with miRNAs, ideal biomarker candidates for AD.

## 4. Targets for Therapeutic Treatment of AD

Increasing our knowledge of the various genetic risk factors and underlying epigenetic mechanisms involved in AD will enable the development of new approaches to therapeutic treatments that will reverse at least some of the cognitive impairment associated with the disease. Research into specific mutations in risk genes and epigenetic marks is also increasing, and the application of effective therapies requires studying of the various cell signaling pathways and, also, regulatory mechanisms by which the targets are altered. In recent years, GWAS studies have focused primarily on specific phenotypes that include the age of onset, differences in ethnicity and psychotic traits in a setting where the epigenome may influence the etiology of AD. However, an effective treatment involves not only the prescription, dosage and proper management of medications but also the implementation and adherence to a set of daily routines consistent with a healthy lifestyle that promotes socialization, nutrition, exercise and mental agility [184].

### 4.1. Targeting Key Genes for AD

Mutations in genes affecting the processing of APP and the formation of Aβ are a potential target for the development of future therapies, such as the S169 del mutation in the familial AD gene *PSEN1* [40]. As mentioned above, BACE2 plays an important role in APP-Aβ processing and cleavage, and evidence suggests that both mutations and *CLU* interference in the APP juxta-membrane helix promote early APP cleavage by suppressing the β-secretase activity [17]. Therefore, the development of targeted therapies that prevent either of these triggers, for example, by preventing *CLU* expression, would help to prevent disease progression. Similarly, the use of drugs that stimulate microtubule stabilization would prevent their dissociation and help reduce MAPT-associated tauopathy in the brain due to mutations or hyperphosphorylation [185]. Researchers have also shown that the processing and cleavage of APP into extracellular vesicles occurs in advanced AD brain cells and contributes to both APP -dependent neurotoxicity and disease prognosis. Surprisingly, at disease onset, the vesicles are released outside the brain along with APP and derived deleterious metabolites such as AICD and Aβ, preventing neurotoxic peptides from accumulating in brain cells [186]. These findings strongly suggest that extracellular vesicles may have a protective function in brain cells, making this mechanism another potential target against AD pathogenesis.

As recently observed in the brains of sporadic AD, the β-secretase-derived C99 fragment participates in the Aβ-independent aggregation of APP-CTFs into mitochondria. This then leads to morphological changes and the functional and homeostatic deterioration of mitochondria that will eventually trigger pathogenic pathways in AD brains. Presumably, therefore, the treatment of these mitochondrial abnormalities would help to counteract the early accumulation of APP-CTFs and slow disease progression [13]. The localization of tau and Aβ, on which their deleterious or potentially beneficial effects depend, must also be considered, so that therapeutic treatments based on these proteins and APP processing must take into account factors that alter their functionality, as well as changes within the cell or their expression in specific cells or body parts. *TREM2* is also becoming an important target in ongoing research to develop AD therapies, with the main goal being to stimulate *TREM2* signaling either early in the disease and before tau neuropathogenesis or before amyloid deposits form in the brain. However, the involvement of *TREM2* in AD needs further investigation, as its effects may be beneficial or detrimental depending on various factors, such as the disease model used and the stage of tauopathy [187].

### 4.2. Epigenomic Biomarkers for the Treatment of AD

Some epigenetic marks may also affect APP processing and Aβ metabolism. In the wake of these discoveries, AD research has recently aimed to highlight specific targets for disease diagnosis and treatment, including noncoding RNAs, in many cases, and various microRNA molecules that hold particular promise for the development of new therapies for neurodegenerative diseases [145,167]. Thus, some types of miRNAs have been shown to interfere with the processing of Aβ and nontoxic APP via the alternative nonamyloidogenic pathway in which soluble APPα is formed by the action of *ADAM10* [188] (Figure 3). Moreover, a recent analysis of several specific miRNA molecules showed that *ADAM10* is regulated by miR-221 in AD neuroblastoma cells and that inhibition of the expression of this microRNA molecule resulted in increased *ADAM10* levels [171]. It is also noteworthy that the mechanism of the small nuclear U1 ribonucleoprotein complex (U1-snRNP) leads to alterations in the neuronal cell cycle via a defective RNA splicing process and ultimately affects the metabolic and biochemical processes responsible for neuroinflammation, cell decay and death [8,189]. Other putative epigenomic biomarkers such as miR-129, which is thought to be ubiquitously upregulated in AD brains, may be useful for drug treatment against target genes [174].

### 4.3. Aβ Immunotherapy

Much attention has been given to the possibilities of Aβ immunotherapy. Although the treatments developed to date have been effective in clearing amyloid plaques in the human brain, they have not been able to slow disease progression or halt cognitive impairment. However, the approach of immune checkpoint blockers (ICBs) now offers a more effective, promising epigenetic target for the development of appropriate therapies against the neuroinflammatory processes that characterize AD amyloid pathology [8]. Some of these are being investigated as therapies for complex diseases such as cancer and are based on antibodies against the programmed cell death system protein ligand-1 complex (PD-L1), a surface receptor for the immune checkpoint inhibition of activated regulatory T cells that has multiple functions related to immune homeostasis. In AD animal models, the use of drugs targeting the PD-1/PD-L1 complex can elicit an immune response that prevents the accumulation of APP-dependent neurotoxicants [190]. A blockade of the PD-L1 complex promotes ligand degradation in antigen-presenting cells by increasing the immune tolerance and preventing T-cell degradation, ultimately helping to reduce inflammation and improve impaired cognitive function [191,192]. On the downside, despite the resources currently being devoted to the development of new BCIs, the results of various experiments are inconsistent in terms of their therapeutic ability to treat AD [193,194], and further research is needed in this area.

### 4.4. HDAC Inhibitors as Therapeutics for AD

Based on the fact that the suppression of HDAC2 and HDAC3 enzymatic activities can promote memory development and learning associated with increased synaptic transmission [195], targeted therapy with HDAC inhibitors (HDACis) aims to reduce the cognitive deficits associated with AD or other neuropathologies [149,196]. To date, both the US and Chinese FDAs have approved some HDACi drugs, such as vorinostat (SAHA), panobinostat (LBH589), belinostat (PXD101), romidepsin (FK-228) and chidamide (HBI-8000), and although most of them were primarily developed to treat hematologic malignancies, some are also being investigated for the treatment of CNS disorders [197]. In addition, lacosamide [198], tubastatin A [199], quisinostat [200], trichostatin A [201] and M344 [202] are the other HDACi that have recently been reported as prominent targets for AD. Valproic acid [203], 4-phenylbutyrate [204], MPT0G211 [205] and nicotinamide [206] also showed similar therapeutic effects in AD animal models. The fusion of the critical structural features of the antioxidant ebselen and HDACi pharmacophores (vorinostat, tubastatin A, panobinostat and quisinostat) served to create a class of novel synthetic hybrid compounds for AD therapy, and the compound identified as 7f was a potent HDACi [207]. The efficacy of the compounds CM-414 and CM-695 as a novel, multitarget therapy focused on the inhibition of HDACs and phosphodiesterase 5 (PDE5) was demonstrated in Tg2576 mice, showing the inhibition of intermediate class I HDACs and a greater inhibition of HDAC6 and PDE5 [208,209]. Finally, Lim et al. pioneered the development of novel aspirin-inspired acetyl donor HDACi [197].

## 5. Concluding Remarks

AD Research has made considerable progress in recent years, particularly in understanding the neuropathological manifestations and etiology of the disease. Concurrent with the advent of new diagnostic tools and improved methods for detecting risk variants, the methods for the quantitative analysis of complex traits have changed in parallel with the development and application of extensive DNA sequencing techniques. However, understanding the exact determinants of this disease remains a lifelong challenge. Recent meta-analysis and global studies of the genome, and in particular the epigenome, have revealed important aspects of the disease that were previously unknown. highlighting the role of epigenetic factors also in the development of age-related cognitive disorders such as dementia [118]. Together with the multitude of bioinformatics resources available, systems biology now enables the integration of epigenomic and genomic data to determine the impact of epigenetic mechanisms in the context of complex disease phenotypes. Indeed, this seems to be a necessary step to better understand the etiology of AD. Fortunately, the discovery of site-specific DNA methylation in the genome, in addition to other relevant epigenetic marks, helps to broaden this view. Limitations to this progress arise from factors related to the experimental design and/or conditions used, as well as causal factors that remain uncontrolled under laboratory conditions or are highly dependent on the targeted effects of specific epigenetic modifications. These include the use of small or nonstandard sample sizes in different experiments or marked tissue specificity, as most analyses focus exclusively on epigenetic changes related to DNA methylation, and thus inadequately capture the fingerprints of variation among different interacting epigenomic compartments. The complexity of neural networks often causes animal models to misrepresent variation in gene expression, while human models struggle with the problem of interindividual variation. Therefore, the poor performance of the models used is also an item on the list of weaknesses to focus on in order to maximise the potential of epigenomic resources. Resolving these conflicts will provide a clearer framework that highlights the mechanistic insights and improves disease management, ultimately contributing to better therapeutic treatment of the disease.

## Figures and Tables

**Figure 1 genes-12-01959-f001:**
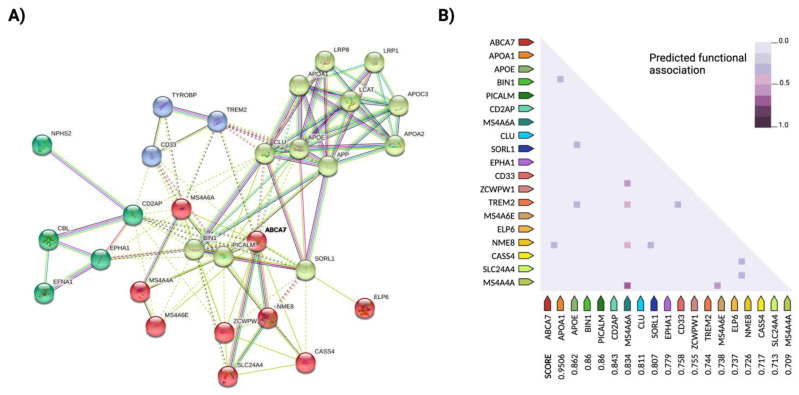
Epistatic interactions between the major susceptibility loci of Alzheimer’s disease (AD). (**A**) Cluster analysis of the protein–protein interaction network of the major susceptibility loci of AD, generated with STRINGdb (http://string-db.org) (accessed on 15 November 2021). The ATP-binding cassette member 7 gene (*ABCA7*) (in bold) was used as the reference protein for the query to identify potential epistatic interactions between all the protein-coding genes. Nodes represent proteins, and edges represent functional and physical protein–protein associations with a significant contribution of the proteins to a common function, regardless of their physical binding to each other. The color of the lines indicates the type of interaction, and the line thickness indicates the strength of the data support. The PPI network analysis was performed with greater than 70% confidence (required minimum interaction score of 0.7). The analysis used a k-means clustering approach with an average local clustering coefficient of 0.729. Four clusters were identified for the network, highlighted in different colors, with dashed lines indicating edges between clusters. The network contains 29 nodes with an average node degree of 7.86; the confidence threshold was set to 0.7 (high). The enrichment of the connectivity in the network was significant (*p* < 0.001), suggesting that the proteins as a group are at least partially biologically connected and have significantly more interactions with each other than expected (114 edges compared to 31 expected edges) from a random group of proteins of the same size and degree distribution from the genome. (**B**) Heatmap of predicted functional associations between search protein *ABCA7* and the other risk loci for AD. Scores refer to the strength of evidence found in a series of experiments for correlated expression between two coding protein genes based on RNA expression patterns and protein coregulation data from ProteomeHD (https://www.proteomehd.net/proteomehd) (accessed on 15 November 2021). Legend: The color intensity indicates confidence in the predicted functional association between *ABCA7* and a given protein (adapted from STRINGdb). Created with BioRender.com.

**Figure 2 genes-12-01959-f002:**
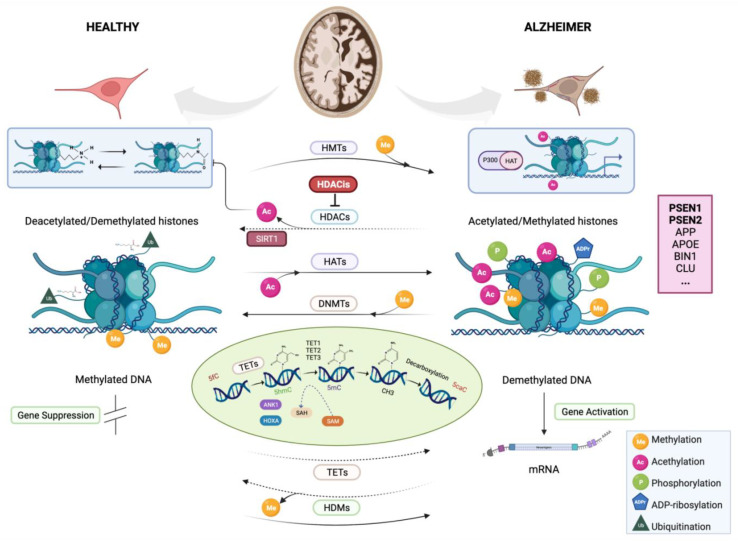
The epigenetic drivers of Alzheimer’s disease (AD). The major mechanisms of epigenetic regulation include DNA methylation and post-translational histone modifications associated with the activation or repression of a gene expression. A number of enzymes catalyze DNA methylation/demethylation reactions, as well as various histone modification processes, such as methylation/demethylation, acetylation/deacetylation, ubiquitination and phosphorylation. DNA methylation/demethylation: DNMT enzymes catalyze DNA methylation by covalently transferring a methyl group from SAM to the 5′-end of cytosine residues (5mC) in CpG islands (CGIs) near gene promoter regions that trigger gene silencing. One widely studied mechanism is DNA hypermethylation in a region of the *HOXA* gene associated with AD neuropathology in the human cortex. In contrast, TETs demethyltransferase enzymes catalyze DNA demethylation by hydroxymethylation from 5mC to 5hmC at CGIs near gene promoter regions, triggering their activation. Histone (Ac) ethylation/deacetylation: covalent binding of an acetyl group to amino acid residues in the histone tail is mediated by HAT enzymes and leads to the regulation of gene activity. The levels of HDAC enzymes such as HDAC2 and HDAC6 tend to be elevated in AD brains. Their deacetylation activities cause gene silencing, while their inhibition helps slow cognitive decline. Histone methylation/demethylation: HMT enzymes methylate histone amino acid residues, thereby inhibiting gene transcription, while HDMT enzymes demethylate them, thereby activating transcription. Ubiquitination of histones: the binding of ubiquitin to lysine residues promotes the proteasomal degradation of tagged proteins, preventing the accumulation of tau and Aβ aggregates, whereas proteasomal inhibition promotes the accumulation of tau and Aβ. Histone Phosphorylation: associated with increased gene expression. Dysregulation of acetylation of non-histone proteins is associated with the pathogenesis of AD. The histone acetyltransferases CBP and p300 are involved in the selective Aβ-induced acetylation of several lysine residues, including those of the transcription factor NF-κB. Mutant variants of *PSEN1* inhibit the degradation of HAT by the proteasome and promote gene expression. The activity of p300 regulates the acetylation of tau by repressing the degradation of p-tau, promoting the microtubule destabilization associated with tauopathy. The histone deacetylase Sirtuin 1 (SIRT1) also regulates NF-κB and is involved in the deacetylation of tau and protection against AD neurodegeneration. Expression of the gene encoding the same enzyme (*SIRT1*) can be altered by the action of p300, leading to hyperacetylation in AD brains. Created with BioRender.com.

**Figure 3 genes-12-01959-f003:**
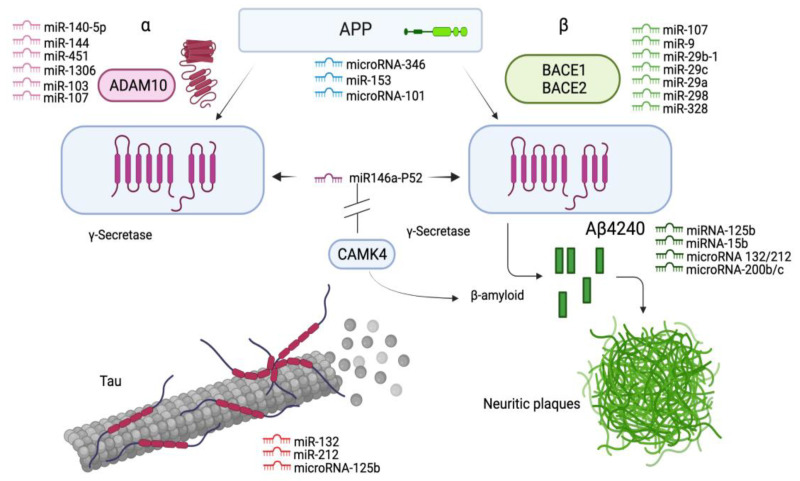
Epigenetic regulatory mechanisms of miRNAs. By regulating the target genes, miRNAs have the ability to alter the processing of amyloidogenic APP into neurotoxic Aβ-42/40 and p-tau aggregates through the sequential action of the cleavage enzymes BACE1 and γ-secretase. Inhibition of the *CAMK4* gene, which is responsible for regulating synaptic functions in neuronal cells, by miRNAs promotes tauopathy and amyloid plaque formation. Similarly, inhibition of the Dicer/Drosha enzyme complex leads to the cessation of miRNA biogenesis and is indirectly associated with the downregulation of DNMT enzymes and, thus, DNA methylation. The *ADAM10* gene is involved in APP processing and Aβ-amyloidosis in AD brains, where its inhibition by specific miRNA molecules leads to overexpression of the gene. Created with BioRender.com.

## Data Availability

Not applicable.

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
