# Peer review of "Advances in Genetics and Epigenetic Alterations in Alzheimer’s Disease: A Notion for Therapeutic Treatment"

_genes, 2021, doi:10.3390/genes12121959_

Round 1

Reviewer 1 Report

This review article titled "Advances in genetics and epigenetic alterations in Alzheimer's disease. A notion for therapeutic treatment" presents a comprehensive discussion of the knowledgebase on the genetic and epigenetic factors linked with the etiology of Alzheimer's disease and its therapeutic targets. Overall the review is insightful and well organized, should address the following minor issues:

  1. Line 141-144: Please clarify whether the APOE allelic frequencies mentioned here are frequencies observed in the general population or specifically among individuals diagnosed with AD.
  2. Neurofibrillary tangles are abbreviated as "NTFs" instead of "NFTs" in lines 82, 89, 92, 99, 120.
  3. Redundant sentences in lines 789-792.

Author Response

RESPONSE TO REVIEWER 1 COMMENTS

This review article titled "Advances in genetics and epigenetic alterations in Alzheimer's disease. A notion for therapeutic treatment" presents a comprehensive discussion of the knowledgebase on the genetic and epigenetic factors linked with the etiology of Alzheimer's disease and its therapeutic targets. Overall the review is insightful and well organized, should address the following minor issues:

We would like to thank you for taking the time to review our manuscript. I will now address each of your comments point by point.

1. Line 141-144: Please clarify whether the APOE allelic frequencies mentioned here are frequencies observed in the general population or specifically among individuals diagnosed with AD.

We have rewritten this paragraph to include a reference that better reflects the proportions of APOE allele frequencies, and we believe it is now consistent with the reviewer's comment.

2. Neurofibrillary tangles are abbreviated as "NTFs" instead of "NFTs" in lines 82, 89, 92, 99, 120.

This has been corrected accordingly.

3. Redundant sentences in lines 789-792.

Removed repeated sentence.

Reviewer 2 Report

The review by Rabaneda-Bueno and colleagues is an excellent and comprehensive review of the current status of our current knowledge of the impact of genetic and epigenetic alteration in Alzheimer’s disease and of high interest for the readership of genes.

Few points need to be addressed.

  1. The language quality, some information, and the comprehension of the Background sections can be significantly improved. The authors are kindly asked to review the whole section and address e.g., the following points:
    1. Line 43: 2-10% of AD cases with onset between 20 and 30 years of age is an inaccurate statement. Please add reference [5] (??) to the sentence “Clinical diagnosis is correct …”
    2. Line 48, the author may add to the statement "postmortem examination is necessary" that nowadays there are a number of tools to facilitate the diagnosis before death, as then explained in detail in the following paragraphs by the authors
    3. Line 74, BACE1 does not generate C83. The authors should review their description of APP processing here and elsewhere, as at times it contains mistakes
    4. Lines 76-80, it is not clear what is meant, it seems to me a contradictory statement
    5. Line 91, the reader may get confused by “plaques form due to pathology” and “as a consequence of”, which appears to me as the same. Maybe is meant “results from an ongoing pathology” versus “contribute to the pathogenic mechanism”
  2. Line 243, it seems obvious that the MAPT gene is instrumental for Tau biogenesis, maybe the authors have something else in mind?
  3. Line 585, not clear what is the link between lysine acetylation in Tau (not the gene!) and plaque formation and APP processing.
  4. Figure 3, gamma-secretase is indicated as delta-secretase. Also, gamma-secretase is a complex of proteins, not just presenilin
  5. Section 4. “Targets for therapeutic treatment of AD”. The authors may want to focus the description of treatments to the advances made in the field of genetic and epigenetic alterations in Alzheimer’s disease, or at list better specify the existing link.
  6. Line 712. There is a contradiction between describing BACE2 as a beta-secretase, defined as the activity responsible for the first cleavage of APP for the generation of Abeta, and stating it does exactly the contrary. BACE2 is not a beta-secretase
  7. The authors may want to review the whole sections and correct some points, examples are:
    1. AD: Alzheimer’s disease
    2. CSB: it is meant CSF?
    3. SRM LC-MS: specify what the abbreviation stands for
    4. BACE2: "beta-secretase precursor" is inaccurate (see also point 6.)
    5. ApoE-E2: specify what the abbreviation stands for
    6. pTMHs: maybe PTM-Hs ?
    7. SUMO: specify what the abbreviation stands for
  8. Table S1. The authors need to review the whole table. In general, instead of “Function” I would use “Proposed function” and instead of “Description” I would use “Hypothetical contribution to disease”
    1. APP: the function of APP is not to regulate APP processing or Notch protein cleavage
    2. PSEN1: does not cleave APP but APP-CTFs
    3. SORL1, same statement is made twice
    4. MAPT, microtubules are made of tubulin, unclear statement
